# Using the Brief Health Literacy Screen in Chronic Care in French Hospital Settings: Content Validity of Patient and Healthcare Professional Reports

**DOI:** 10.3390/ijerph18010096

**Published:** 2020-12-25

**Authors:** Adèle Perrin, Luiza Siqueira do Prado, Amélie Duché, Anne-Marie Schott, Alexandra L. Dima, Julie Haesebaert

**Affiliations:** 1Health Services and Performance Research, Université Claude Bernard Lyon 1, 69100 Lyon, France; adele.perrin@univ-lyon1.fr (A.P.); luiza.siqueira-do-prado@univ-lyon1.fr (L.S.d.P.); amelie.duche@etu.univ-lyon1.fr (A.D.); anne-marie.schott-pethelaz@chu-lyon.fr (A.-M.S.); alexandra.dima@univ-lyon1.fr (A.L.D.); 2Pôle de Santé Publique, Hospices Civils de Lyon, 69002 Lyon, France

**Keywords:** health literacy, chronic care, content validity, BHLS

## Abstract

Person-centered care has led healthcare professionals (HCPs) to be more attentive to patients’ ability to understand and apply health-related information, especially those with chronic conditions. The concept of health literacy (HL) is essential in understanding patients’ needs in routine care, but its measurement is still controversial, and few tools are validated in French. We therefore considered the brief health literacy screen (BHLS) for assessing patient-reported HL in chronic care settings, and also developed an HCP-reported version of the BHLS with the aim of using it as a research instrument to assess HCPs’ evaluation of patients’ HL levels. We assessed the content validity of the French translation of both the patient-reported and HCP-reported BHLS in chronic care within hospital settings, through cognitive interviews with patients and HCPs. We performed qualitative analysis on interview data using the survey response Tourangeau model. Our results show that the BHLS is easy and quick to administer, but some terms need to be adapted to the French chronic care settings. Health-related information was observed to be mainly communicated orally, hence a useful direction for future literacy measures would be to also address verbal HL.

## 1. Introduction

Limited health literacy (HL) is associated with worse health outcomes in general [1]. According to the World Health Organization (WHO), HL is defined as “the cognitive and social skills that determine the motivation and ability of individuals to gain access to, understand and use information in ways that promote and maintain good health” [2]. For people living with chronic conditions, HL is particularly important [1] since managing chronic conditions involves interacting with illness-related information such as diagnoses, medical treatments, or lifestyle changes required, as well as interacting with healthcare professionals (HCPs) and the healthcare system. From HCPs point of view, care has shifted from professional- or facility-centered medicine to person-centered, i.e., care that is respectful of and responsive to preferences, needs, and values of individuals, and ensures that values of individuals guide all decisions [3]. Shared decision-making is thus being put at the center of the consultation [4] to promote person-centered care. In this context, HCPs must be able to detect patients’ HL needs to improve communication and allow effective patient participation in the care process [5,6]. Therefore, HL has emerged as an important field of research in chronic care [7]. Its assessment becomes increasingly advocated to support services and HCPs adapting written and oral communication to individual needs [8,9].

However, assessing HL in routine care is a complex issue, as many definitions and measurement instruments are available; it may be difficult to know which one to apply to the broad range of situations encountered, especially in chronic care [10]. At present, there is no standard instrument to be used in current practice and HCPs adapt communication to their subjective perception of patients’ understanding. Nutbeam et al. [11] proposed three dimensions for HL: functional (the ability to read and understand written information), interactive (more advanced cognitive literacy and social skills that enable active participation in health care), and critical (the ability to critically analyze and use information to participate in actions that overcome structural barriers to health). Available instruments align different definitions of HL covering one or more of these dimensions [10], measure either objective (performance-based) or subjective (self-perceived) HL, vary in length and administration mode. While lengthy multidimensional and performance-based measures are often preferred for research purposes, short subjective measures are more adapted to the time constraints of routine care. One such measure is the brief health literacy screen (BHLS), a 3-item questionnaire with response options based on a 5-point Likert scale that takes approximately 1 min to complete and is either self- or orally administrated. It was developed in 2004 by Chew and others to measure functional HL [12], validated in English and applied in different contexts [9,10,11,12,13,14,15]. Its performance for detecting low levels of functional HL is similar to that of the rapid estimate of adult literacy in medicine (REALM) [16] and the short test of functional health literacy in adults (S-TOFHLA) [17], which have been used in the large majority of empirical HL research [10] and are highly correlated with each other [12,15,18]. In addition, to our knowledge, there is no instrument dedicated to HCPs to assess how they perceive patients’ HL levels. As HCPs perceptions of patients’ HL level during clinical interactions are instrumental for appropriate tailoring of communication, this approach could be convenient, especially for research purposes, to evaluate whether HCPs correctly appraise patients’ HL levels during patient-HCP interactions.

In France, the use of the BHLS has been limited so far to online surveys [19,20]. Although a version in French has been produced by Delanoë et al. [21] for use in Quebec (Canada), currently there is no standard validated version to be applied in clinical and hospital settings in France. Such a version would be useful both for research and for routine use to assess patients’ HL and adapt interventions. For these purposes, it is necessary to continue the psychometric validation process following current standards, i.e., to investigate whether the French translation meets the three key criteria of content validity-relevance, comprehensiveness, and comprehensibility as recommended by the COSMIN (COnsensus-based Standards for the selection of health Measurement INstruments) guidelines [22,23].

Thus, the objective of this study is to perform content validity testing for patient-reported and HCP-reported versions of the BHLS in chronic care within French hospital settings, through cognitive interviews. Assessing to what extent the construct of HL as defined by the BHLS corresponded to the way patients and HCPs interpreted the questions would allow a more nuanced understanding of the suitability and potential uses of this instrument and identify ways of improving measurement of HL in these settings.

## 2. Materials and Methods

We conducted a qualitative study using semi-structured cognitive interviews, a method to investigate if a survey question corresponds to its intended purpose, to assess the content validity of the French version of the BHLS in chronic care within hospital settings. This study followed the COSMIN Guidelines [23] to test if the questionnaire measurement was an adequate reflection of the construct to be measured (HL). The interview guides followed the 4-stage Tourangeau model [24] as theoretical basis; this model states that survey responses involve the following four stages: (1) comprehension of the terms/questions, (2) retrieval of relevant information from memory, (3) judgement/estimation processes, and (4) response processes regarding answer modalities [25]. The BHLS in English is presented in Table 1.

Following recommended best practices, all participants were informed orally on the study objectives and their oral consent was collected before inclusion. In accordance with French legislation, study-specific ethical approval was not required as due to the nature of the study objectives, i.e., understanding how participants perceived a questionnaire, and not directly improving their own health and care. No identifiable data were collected, and data processing followed recommended procedures at the research site.

### 2.1. Development of the BHLS in French

The first version of the BHLS in French (Appendix A) was developed by Delanoë et al. [21] as a part of a previous work conducted with French-speaking patients (patient-reported BHLS V1). It was designed using a forward-backward translation procedure (from English to French to English). As required by COSMIN guidelines [23], two translators produced a first version of the English text in French, then other two translators produced a new version of the translated text in English, which was compared to the original text. Consensus on questions wording was then reached after revision by a committee composed of researchers, healthcare professionals, and translators.

To assess the content validity of the French patient-reported BHLS (V1) in hospital settings, we conducted interviews with a sample of patients living with chronic conditions. A second version of the patient-reported BHLS was produced after the analysis of patients’ interviews results (V2). After that, a version based on the patient-reported BHLS V2 dedicated to interviewing HCPs was created by a group of experts in HL, public health, and social psychology. We then performed the content validity assessment of the resulting HCP-reported BHLS V1 through interviews with HCPs (Figure 1).

### 2.2. Recruitment of Participants

Patients were recruited in May 2019 using convenience sampling at a respiratory diseases department at the university hospital Hospices Civils de Lyon (HCL), France. Inclusion criteria were: (1) being at least 18 years old, (2) having an appointment in the outpatient clinic of the participating hospital for the respiratory follow-up of a chronic disease during the study period, (3) being able to give consent, and (4) being able to communicate in French. 

HCPs were recruited between January and June 2020 in the departments of cardiology and psychiatry of the HCL and diabetology of the Centre Hospitalier Universitaire Grenoble-Alpes (CHUGA). The inclusion criterion was to work in a hospital facility in chronic care. Sampling aimed for a diversity of professions: physicians, nurses, and assistant nurses. 

We aimed to recruit at least 12 participants in each group (patients and HCPs), expecting to reach data saturation with this sample size following recommendations on qualitative interviews [26].

### 2.3. Interview Methods

The BHLS was self-administered in a separate room (patient-reported BHLS V1 and HCP-reported BHLS V1) and the cognitive interview was undertaken. Three investigators (A.P., L.S.d.P., and A.D.) led the interviews after being trained in interviewing methods and having practiced with the interview guide with members of the research team. The interviewer asked a series of questions to obtain specific information on the process by which the participant answered each BHLS question [25]. 

We designed two interview guides, one for patients and one for HCPs. Patients were invited to think about different experiences throughout the care process, and HCPs were prompted to recall a patient living with a chronic condition who had recently been in care. These guides investigated the reasoning that leads to answering the questions and identified problematic items to subsequently drive the necessary modifications to enhance content validity [27]. The interview guide for patients’ interviews (summary presented in Table 2) was drafted first, the HCP guide derived from the patient guide after improvement and adaptations to fit with the profiles of HCPs. For the first question, HCPs were asked how confident they believed the patient was to fill in forms by him/herself. We were interested not only in how they perceived the terms, but also in how they perceived patients would fill in forms by themselves. In the second question, we were interested in understanding how HCPs observed and described the frequency of aid requested by patients to read hospital or health-related documents. In the last question, professionals were asked how often they believed the patient had problems learning about their medical condition because of difficulty reading hospital materials.

### 2.4. Qualitative Analysis

All interviews were recorded with participants’ permission and transcribed in summary tables for qualitative analysis. The summary table template followed the 4-stage Tourangeau model [25]. Analyses were performed following recommended procedures for cognitive interviews [25,28] and were conducted separately for patients and HCPs. 

First, two independent coders (A.P. and A.D. for patients; A.P. and L.S.d.P. for HCPs) extracted information from the recordings to fill in the summary tables, then quotes were compared to define categories from similar responses to establish an analysis codebook (each code was followed by a code definition and instructions on how to code); it was then tested by both coders in one interview for each group; categories were extensively discussed and later reviewed by A.L.D. and J.H. Each coder then applied the codebook to half the interviews and coders reviewed all the summary tables together. Two synthesis tables, one for patients and one for HCPs (Appendix A), were generated from the summary tables; key findings were analyzed to determine how participants had understood the terms in the questions and if the elements and items were interpreted as intended by the instrument. 

## 3. Results

We conducted cognitive interviews with 13 patients attending outpatient consultations for a chronic disease and 12 HCPs to assess content validity of the French version of the BHLS. For both groups, the number of participants was sufficient to reach data saturation [26,28].

### 3.1. Patients Results

Eight of the interviewed patients were men. Over half of them were between 60 and 69 years old (*n* = 7) and 6 had a high level of education. There were as many active (*n* = 6) as retired patients (*n* = 6). None self-reported a low level of health literacy (Table 3). 

Results are reported according to the stages of Tourangeau’s model for each of the three questions (Q1, Q2, and Q3) of the patient-reported BHLS V1.

#### 3.1.1. Comprehension of the Questions and Terms

In Q1, the term “forms” was interpreted in two ways; either it was associated with administrative forms (e.g., taxes or administrative forms in English, “impôts” or “formulaires administratifs” in French) (*n* = 7), or it was understood as medical forms (*n* = 6). Eight of the thirteen respondents thought the question was about questionnaires. When asked about the meaning of “to be confident,” in the French translation “être confiant,” respondents mainly referred to being self-confident (*n* = 8) but it was sometimes seen as trusting the doctor or the people administering the questionnaire (*n* = 5).

Regarding Q2, most participants (*n* = 12) considered that “have someone help you” (“demander de l’aide” in French), was related to asking for HCPs help, while two patients referred to their relatives. Ten patients associated “hospital materials” (“documentation remise à l’hôpital” in French), to written materials and two patients also considered oral information. Furthermore, the majority linked it to documentation related to illness and care (results of biological or radiological exams) while the others thought it would refer to more general medical documentation (information leaflets).

Regarding Q3, five patients interpreted “have problems learning” as having difficulty understanding terms or words employed in health documents that they considered too technical or specific. Two patients pointed out that in general health documents or information leaflets were incomplete in order to fully understand the information. The term “medical condition” was mainly associated with health status (“état de santé,” in French) but three patients also referred to “medical record,” “treatment or care” or even “comprehensive support.” This expression was considered too vague. Complex medical terms were unanimously interpreted as terms with which they have “trouble understanding,” but two patients also talked about the concept of readability of manuscript documents, i.e., “the doctor’s handwriting.”

To improve comprehension and adapt to cultural specificities, in the first question the term “form” was replaced by “medical form” (“formulaire médical”) and in the third question the expression “medical condition” was replaced by “health status” (“état de santé”), according to suggestions made by the respondents.

#### 3.1.2. Retrieval of Relevant Information from Memory

All participants referred to outpatient consultations at the hospital to fill in the BHLS, and especially the consultation of the day just before the interview. However, regarding the term “forms,” some patients referred to other situations of their daily life, unrelated to health and care.

#### 3.1.3. Judgment/Estimation Processes

With regard to the decision-making process for the entire questionnaire, respondents declared that they did not have to make a considerable effort to recall personal situations. Their answers were intuitive and spontaneous. In addition, they indicated that they did not feel influenced or even guided in their choice of response.

Q1 made good reference for most patients (*n* = 8), to situations experienced in consultation, concerning treatments or forms to be filled in for the pathology. Although the interpretation of the term “form” itself remains confusing.

Q2 mostly referred to situations of care at the hospital and the examinations to be carried out. However, almost all patients referred to hospital HCPs and did not think about their relatives (spouse, child, neighbor) or other HCPs when approaching the subject asking for help.

Lastly, for Q3, the patients referred to situations related to the management of their chronic pathology. Most of them referred to written documents because of the complexity of the terms or the difficulty in reading the doctors’ writing. On the other hand, one patient understood the question in terms of oral comprehension. The little variability in response meant that the subjects were considered to have had similar interpretations among themselves and in accordance with what the question required.

#### 3.1.4. Response Processes Regarding Answer Modalities

For Q1, answers modalities were from “extremely confident” to “not at all confident.” Only the first two response modes (quite a bit confident, extremely confident) were given. “Extremely confident” was interpreted on the one hand as being entirely sure (“flawless,” “ten on a scale of ten”) and on the other hand, as quite sure (“90%,” “possible problem but rare”). While “somewhat confident” was explained as “not totally sure” or “sometimes a doubt.”

Q2 and Q3 asked about frequency of the situation (never, occasionally, sometimes, often, always). The answer mode “never” was defined as “0 times” or “a few times” while the answer mode “occasionally” implied “from time to time” or “infrequently.” “Sometimes” was also explained as “from time to time.” Answers modalities were sometimes explained in the same way although they were supposed to express nuances, which shows variability between individuals’ comprehension. 

The BHLS was judged as easy to complete by almost half of the respondents (*n* = 5), while other 5 respondents considered the last question hard to understand.

### 3.2. HCPs Results

Seven HCPs were women, a third were between 40 and 49 years old and a third were trained physicians (Table 4).

#### 3.2.1. Comprehension of the Questions

Regarding Q1, “to be confident” was understood in two different ways: to be sure and to be able to fill in the forms (*n* = 10), and to trust the professionals and institutions they would hand over the forms to (*n* = 2). All HCPs understood the term “medical forms,” but four of them said it was not used in their work setting. When asked for examples of medical forms, HCPs mentioned consent forms for research or invasive procedures (*n* = 6), medical history forms (*n* = 2), hospital admittance/discharge forms (*n* = 4), questionnaires for research studies (*n* = 1), satisfaction forms (*n* = 1), insurance forms (*n* = 1), and suicidal risk forms (*n* = 1). Three HCPs also cited prescriptions, although they are filled in by physicians, and two did not provide any example. 

Regarding Q2, for seven HCPs “have someone help” meant the patient would ask questions orally to friends or family, while two referred to asking questions orally to HCPs, and two referred to asking questions both to friends or family and HCPs. Concerning the help provided, HCPs said it meant asking questions about the documents (*n* = 7), health care (*n* = 1), or the disease itself (*n* = 2), one said it meant calling the hospital service after discharge to ask for help understanding the course of treatment. One HCP did not give a clear answer. When asked how they understood hospital or health-related documents, five referred to documents related to health care or the specific disease the patient had, two referred to general documents delivered by the hospital to all in-patients, and five said either general or disease-specific documents. 

Regarding Q3, “problems learning” was perceived as a lack of general literacy by one HCP and “difficulty reading” was perceived likewise by five HCPs. For the remaining HCPs, both expressions were perceived as related to the complexity of medical terminology in the delivered documents. Eight HCPs understood that “health status” was about the disease itself, while three of them understood it meant the health care the patient had undertaken. One HCP did not give a clear answer. 

#### 3.2.2. Retrieval of Relevant Information from Memory

For Q1, only three HCPs provided a description of an actual situation they had experienced. Nine said they had not experienced any situation where the patient had to fill in forms, so they recalled mentally possible forms patients could be asked to fill in or projected onto the specific patient other experiences they generally had with other patients, mainly related to difficulties reading documents or understanding information given orally. For Q2, five HCPs were able to recall a real situation and six for Q3. 

#### 3.2.3. Judgment/Estimation Processes

To answer Q1, most HCPs assessed the patient’s confidence based on the situations not related to filling in medical forms, e.g., having seen them reading documents, or based on their knowledge of the patient’s educational background. 

For Q2, HCPs answered based on how patients asked them for help, or on how they had seen patients interact with their families. The phrase “at what frequency” (“à quelle fréquence”), translation of “how often,” was perceived as too vague. Q3 was the most challenging, many said they did not relate patients’ difficulties apprehending a specific medical condition to problems reading, instead they said it was mainly related to the disease complexity or to a substantial lack of motivation (which they often called “denial,” “déni” in French). 

Eight HCPs considered the questionnaire easy to answer, three said it was difficult, and one said the third question was long and confusing. Many said they were aware that their answers may not reflect the patient’s reality, but their perception of it. 

#### 3.2.4. Response Processes

For Q1, HCPs’ answers varied more than patients. Five HCPs chose the response option “quite a bit,” four “somewhat,” two “not at all,” and one “a little bit.” “Quite a bit” was mainly associated with perceived higher educational levels, while “somewhat” was associated with patients who were able to cope but had doubts or forgot important information. One HCP stated to have chosen “somewhat” because it appeared to be more neutral than the others. “A little” was associated with a recent diagnosis and “not at all” with patients that needed help reading. 

For Q2, three HCPs chose the option “a little of the time,” three chose “most of the time,” and “none of the time,” “some of the time” and “all of the time” were each chosen by two HCPs. One HCP explained having chosen the “a little of the time” because “every time we saw the patient, he had additional questions to ask.” “Most of the time” was associated with patients that asked questions repeatedly. “None of the time” was associated with patients who had higher educational levels, “some of the time” with a perception that the patient could have problems understanding documents or the choice was based on the perception of the response option being neutral, and the “all of the time” was associated with patients with a low autonomy degree, e.g., they generally have a close relative present during consultations. 

For Q3, four HCPs chose the option “most of the time” and it was associated with patients that had problems understanding written documents but did not have problems systematically; three chose “a little of the time,” which was associated with patients that understood correctly but could have doubts. “None of the time” was chosen by three HCPs and “some of the time” by two, and both were associated with the same characteristics as for the second question (the first to higher educational levels and the latter to a perception of neutrality). 

Table 5 presents the key findings for both groups. 

## 4. Discussion

The BHLS was translated to French, yet no validation has been performed specifically for use in hospital settings. In addition, little is known about how professionals perceive and evaluate patients’ literacy levels to adapt their communication in clinical interactions, and whether the BHLS could be used to measure HCP appraisal of patients’ HL. Therefore, in this study we have developed an HCP-report version of the instrument and examined the content validity of the French version of the BHLS for patients and the HCPs version introduced here, in chronic care within hospital settings in France through cognitive interviews and qualitative analysis of patients and HCPs reports. Our findings suggest several uses are possible for the BHLS in French hospital settings. According to participants in our study, the BHLS was overall quick and easy to administer, which supported its applicability in clinical practice. However, we also identified limitations to its validity in the studied context, especially related to the terms used in the questions, which highlight several directions for further improvement. 

HL is an evolving concept, changing with factors internal and external to health care systems. Therefore, measurement instruments like the BHLS need to be accompanied by a continuous process of psychometric evaluation and improvement to suit the needs of both patients and HCPs in current practice. Some limitations identified in the current versions are likely due to cultural or healthcare system-specific factors such as terms culturally adapted to the United States [12,13,15,18,29] that were not perceived the same way in France. For example, participants struggled with connecting the term “forms” with health-related situations and associated it rather with public administration. This may be due to the fact that the healthcare system in France does not systematically require patients to fill in forms by themselves, as most information is obtained orally by HCPs and procedures vary greatly from service to service. Therefore, it would be useful to add to the BHLS questions about verbal HL, i.e., the ability to understand and act on verbally communicated health information [15]. Sand-Jecklin and Coyle [15] have added two questions addressing the understanding of verbal health information to the three screening questions proposed by Chew et al. [12], which might be a starting point for further studies to investigate whether this dimension of HL can prove more adapted to the French healthcare system. A second example refers to the term “confident”; while in English the meaning is unambiguously referring to the respondent’s perceived self-efficacy, in French the term also denotes “trust” and therefore was sometimes interpreted in this context as trust in the patient-provider relationship. A more appropriate term in French would be equivalent to “capable” (“se sentir capable”), which can be tested in further work. A third example refers to Q1 and Q3 being perceived as the most difficult to understand in our study. This finding contrasts to previous results supporting the validity of these questions and indicating strong correlations between the English BHLS and S-TOFHLA, a widely used measure of functional HL [12,30]. Although the BHLS has been extensively validated in the United States [12,13,15,30], in other languages it has not always shown similar results. Mantwill et al. [31], showed that in five countries (Hungary, Italy, Lebanon, Switzerland, and Turkey), the BHLS items were not related with the S-TOFHLA and argued that qualitative studies should be performed outside the United States to investigate cross-cultural differences.

Rapid screening of patients with potential difficulties to understand their medical condition is essential to adapt communication and possibly improve adherence to treatments and self-care behaviors. Participating HCPs were equally positive about the potential application of BHLS in routine care but were, in most cases, aware their answers corresponded to their own perceptions and likely not the patient’s estimation of their own HL level. This shows the HCP-reported BHLS could be useful in further studies, to compare patients’ self-perceived HL levels against HCPs’ perceptions, if applied to the same patient population. Another useful application of HCP-reported BHLS could be for training purposes, to raise awareness on the importance of HL and its assessment to adapt communication. However, given the HCPs expressed concern with potential mismatch between their perceptions and patients’ own assessments, we would not recommend its use as a proxy measure of patients’ HL level until further improvements and testing are performed. To expand the toolbox for assessing HL in clinical settings, other questionnaires such as the recently developed Conversational Health Literacy Assessment Tool (CHAT, [32]) need to be adapted to French and could help HCPs better understand patients’ HL challenges and resources and thus make health care decisions more effectively and responsively. 

The concept of HL is bound to evolve with future changes in healthcare systems. Currently, there is increasing attention to HL in France, and we expect that with further training of HCPs on identifying HL-related patient needs and strengths in the consultation HCPs will feel more equipped to assess HL levels accurately and intervene appropriately. It can be envisaged in this case that the potential mismatch between patient-reported and HCP-reported BHLS would reduce, which could be an indicator of successful HCP training in HL. To increase content validity and also concordance between HCP and patient reports, the evolution of HL measurement would also need to follow changes in how consultations take place, in particular what common experiences of obtaining, processing and understanding information occur in patient-HCP interaction that can be indicative of patients’ HL. Moreover, in France, as worldwide, information is obtained more and more from the internet or mHealth apps instead of information brochures or hospital materials. Therefore, an important dimension to consider for future development of short HL assessments for use in clinical care is digital HL, i.e., the ability to seek, find, understand, and appraise health information from electronic sources and apply the knowledge gained to addressing or solving a health problem [27]. 

This study has some limitations. First, the patient population was a convenience sample recruited in a single hospital department, which can impact the generalizability of our results. To strengthen the results presented here, further studies can recruit patients from different hospital departments to have a more diverse population. Second, we noticed that HCPs often based their answers on situations that they had not experienced with the patient. This may be due to recall bias, since they were not interviewed immediately after a patient visit; however, an alternative explanation may be that situations evoked in the BHLS less frequently occur in HCP-patient interactions in these settings. Third, we interviewed patients after outpatient visits but not after hospitalizations. We believe that this might have led to differences in the type of situations patients and HCPs recalled to answer the BHLS questions, particularly in relation to the term “hospital materials” which were associated with a greater diversity of situations for patients compared to HCPs. One way to address these points would be to recruit both patients and HCPs in the same hospital departments during hospitalization. Finally, we believe the patient-reported BHLS V2, proposed here following the content validity examination, is yet to be tested using qualitative methods with a new patient population. 

## 5. Conclusions

This study adds to the understanding of how patients and HCPs go through the BHLS response process to assess patients’ HL levels. The questionnaire was well-received by participants, and the interviews conducted led to further improvement of the patient-reported BHLS and propose several improvements of the HCP-reported BHLS. The present findings converge with previous work indicating that HL is not an easy translatable concept across cultures and healthcare systems. Interactions between patients and HCPs or health systems may manifest in different ways and require different indicators in different settings. Therefore, further work could be focused on identifying the main categories of interactions in a specific setting and formulate items adapted to the setting that target these categories. We also show that the concept of HL needs to be further discussed in HCP initial and continuous education, so that HCPs develop their skills to identify barriers in patients’ comprehension and develop strategies to enable efficient communication.

## Figures and Tables

**Figure 1 ijerph-18-00096-f001:**
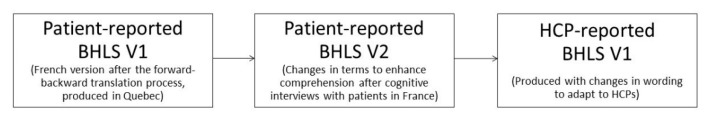
Development of the different versions of the BHLS used in this study.

**Table 1 ijerph-18-00096-t001:** Questions and response options for the brief health literacy screen (BHLS).

Question	Response Options
1. How confident are you filling forms by yourself?	Not at all confident, a little bit confident, somewhat confident, quite a bit confident, extremely confident
2. How often do you have someone help you read hospital materials?	Never, occasionally, sometimes, often, always
3. How often do you have problems learning about your medical conditions because of difficulty reading hospital materials?	Never, occasionally, sometimes, often, always

**Table 2 ijerph-18-00096-t002:** Summary of the interview guide for patients.

Stage	Topic
Comprehension of the terms/questions	
Question 1	Understanding of the term “forms”
	Suggestions to replace the term “forms”
	Understanding of the expression “to be confident”
Question 2	Understanding of the expression “to have someone help you”
	Understanding of the term “hospital materials”
Question 3	Understanding of the expression “problems learning”
	Understanding of the expression “medical conditions”
	Understanding of the expression “difficulty reading”
Retrieval of relevant information from memory	
Question 1	Last time they recalled having filled in forms
	Examples of forms
Question 2	Situation(s) used to estimate frequency of having someone help them
Question 3	Situation(s) recalled appraising difficulties learning about medical conditions
Judgment/estimation processes	
All 3 questions	Effort made to recall situations experienced
	Feeling influenced in replying to the questionnaire
	Feeling a “right” or “wrong” answer was expected
Response processes regarding answer modalities	
Question 1	Meaning of each response option
	Reason to have chosen a specific response option
Questions 2 & 3	How many times having had someone help them or having had problems learning because of difficulty reading
General questions	Understanding of the topic addressed by the questionnaire
	Assessing difficulties to answer to the questions presented
	Opinion on the title and presentation of the questionnaire

**Table 3 ijerph-18-00096-t003:** Patients characteristics.

Characteristic	*n* (%)
Sex, *n* (%)	
Male	8 (61.5)
Female	5 (38.5)
Age in years, *n* (%)	
30–39	1 (7.7)
40–49	2 (15.4)
50–59	2 (15.4)
60–69	7 (53.8)
>75	1 (7.7)
Education level, *n* (%)	
No school	2 (15.4)
Middle school	3 (23.1)
High school	2 (15.4)
College/University	6 (46.2)
Professional status, *n* (%)	
Unemployed	1 (7.7)
Employed	6 (46.2)
Retired	6 (46.2)
Medical condition, *n* (%)	
Asthma	6 (46.2)
Leukemia	3 (23.1)
Spinal cord transplant	1 (7.7)
Allergy	1 (7.7)
Unknown	2 (15.4)
Health literacy level, *n* (%)	
Low (BHLS score ≤9)	0
Adequate	13 (100)

**Table 4 ijerph-18-00096-t004:** HCP characteristics.

Characteristic	*n* (%)
Sex, *n* (%)	
Male	5 (41.7)
Female	7 (58.3)
Age in years, *n* (%)	
20–29	1 (8.3)
30–39	5 (41.7)
40–49	4 (33.3)
50–59	2 (16.7)
Profession, *n* (%)	
Physician	4 (33.3)
Intern	3 (25)
Nurse	3 (25)
Assistant nurse	2 (16.7)

**Table 5 ijerph-18-00096-t005:** Key findings from cognitive interviews following the Tourangeau survey response model.

Respondent	Question	Comprehension	Retrieval	Judgement	Response
Patients	Q1	“To be confident” not always perceived as self-confidence “Forms” is a too broad term	Situations with administrative forms not related to healthcare	Responded spontaneously	The first two response options were chosen by different respondents giving the same justification for their choices
	Q2	“Have someone help you” perceived as mainly related to HCPs help “Hospital materials” perceived as imprecise and referring to general documentation about disease or care	Outpatient consultations		“Never” perceived as not once to a few times “Sometimes” explained as from time to time
	Q3	“Have problems learning” perceived as mainly referring to medical jargon “Medical condition” not always perceived as health status “Trouble understanding” perceived as referring to misunderstandings between patient and HCPs			
HCPs	Q1	“To be confident” not always perceived as self-confidence “Medical forms” described as a rarely used term	Imagined situations	Judged mainly based on educational background	“Quite a bit” for higher education levels “Somewhat” for patients who missed information or to indicate a neutral response “A little” for recent diagnosis “Not at all” for help needed with reading
	Q2	“Have someone help” perceived as mainly related to family or friends “Hospital materials” was perceived as imprecise and referring to general documentation about disease or care	Outpatient consultations or hospitalization	Judged based on frequency of asking for HCPs or family for help	“None of the time” for higher educational levels “A little of the time” for high autonomy, but also explained as every time “Some of the time” for problems understanding documents or to indicate a neutral response “Most of the time” for patients who repeatedly asked questions “All of the time” for patients with low autonomy
	Q3	“Problems learning” perceived as related to medical jargon or to an important lack of motivation “Health status,” term replacing “medical condition,” understood as denoting the disease or care		Judged based on disease acceptation	

## Data Availability

The data presented in this study are available in Appendix A.

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
