# Peer review of "Using the Brief Health Literacy Screen in Chronic Care in French Hospital Settings: Content Validity of Patient and Healthcare Professional Reports"

_ijerph, 2020, doi:10.3390/ijerph18010096_

Round 1

Reviewer 1 Report

Thank you for allowing me to review the article entitled "Using the Brief Health Literacy Screen in chronic care in French hospital settings: content validity of patient and healthcare professional reports," the topic of which I find as interesting as it is relevant to clinical practice.

I have some questions about the paper:

Introduction:

The authors refer to the importance of Health Literacy in chronic pathologies. Do you think it is only necessary to measure Health Literacy in patients with chronic pathologies?

In the introduction it would be good to define person-centered care and joint decision-making as an important part of the therapeutic process in order to understand why health literacy is a necessary process.

If there are many definitions for HL, I would like to see at least one in this introduction. If there are many instruments for measuring LH available, the authors could describe them in order to understand their differences, why there is no standard one and why the Brief Health Literacy Screen is the one chosen for validation

Lines 54 and 55 are missing references for the measurement instruments: Rapid Estimate of Adult Literacy in Medicine (REALM) and the short Test of Functional Health Literacy in Adults (S-TOFHLA),

Materials and methods

I am not aware of the legislation at ¨France, but in other countries an ethics committee should be passed to carry out ta study where patients are audio recorded and socio-demographic data is collected. Make sure of this issue.

Lines 96-97: the back translation was compared with the original. This is the proper procedure, but it should be compared by a team composed of methodologists, health professionals, language professionals and translators who have participated in the process up to this point. https://pubmed.ncbi.nlm.nih.gov/11124735/ Was this the case? Please describe it

Lines 117-118: it is not clear to me why the authors "expected to recruit at least 12 participants in each group (patients and HCPs) to reach data saturation in qualitative interviews," and the article cited does not help me either.

Lines 143-144: Tourangeau's model was followed, which consists of 4 steps, which should be described.

Line 145 “we defined cathegories from similar responres to stablish an analysis codebook”. Who and how many made that category definition? was it done by triangulation? I think that the procedure for extracting and coding the information should be better defined and cited.

Discussion:

I think the discussion should start with the objective of the study and the main results, more or less what is described in lines 303-308, and then be able to discuss them.

I would like to see a discussion about the results of the study, rather than the possible uses of the tool, this is also reflected in the conclusions, where the problems of the construct and its future lines are discussed, instead of indicating the main findings of the research.

Reviewer 2 Report

This study is related to the validity of BHLS and its re-development in French, and the study design method and process were very interesting. Each part in study are described in detail. However, there are some simple suggestions, and I hope to make corrections at the author's decision.1. How about adding “chronic care” in the keyword part?

  1. I think it would be good to use another word to start with "we" in a sentence. There are too many.
  2. 2.2. in the Recruitment of participants

“We expected to recruit at least 12 participants in each group (patients and HCPs) to reach data saturation in qualitative interviews [19].” However, as a result of the study, there were 13 patients, so please check again.

    4 . The first paragraph of the discussion is not clear, so can I delete a few lines?  

  1. Conclusion In the first paragraph, it is necessary to briefly describe the research results

Round 2

Reviewer 1 Report

I would like to thank the authors for their consideration of my comments and the work done to improve the article.

Congratulations